# TOWARDS DEEP INTERPRETABILITY (MUS-ROVER II): LEARNING HIERARCHICAL REPRESENTATIONS OF TONAL MUSIC

**Haizi Yu**
Department of Computer Science
University of Illinois at Urbana-Champaign
Urbana, IL 61801, USA
`haiziyu7@illinois.edu`

**Lav R. Varshney**
Department of Electrical and Computer Engineering
University of Illinois at Urbana-Champaign
Urbana, IL 61801, USA
`varshney@illinois.edu`

## ABSTRACT

Music theory studies the regularity of patterns in music to capture concepts underlying music styles and composers' decisions. This paper continues the study of building *automatic theorists* (rovers) to learn and represent music concepts that lead to human interpretable knowledge and further lead to materials for educating people. Our previous work took a first step in algorithmic concept learning of tonal music, studying high-level representations (concepts) of symbolic music (scores) and extracting interpretable rules for composition. This paper further studies the representation *hierarchy* through the learning process, and supports *adaptive* 2D memory selection in the resulting language model. This leads to a deeper-level interpretability that expands from individual rules to a dynamic system of rules, making the entire rule learning process more cognitive. The outcome is a new rover, MUS-ROVER II, trained on Bach's chorales, which outputs customizable syllabi for learning compositional rules. We demonstrate comparable results to our music pedagogy, while also presenting the differences and variations. In addition, we point out the rover's potential usages in style recognition and synthesis, as well as applications beyond music.

## 1 INTRODUCTION

Forming hierarchical concepts from low-level observations is key to knowledge discovery. In the field of artificial neural networks, deep architectures are employed for machine learning tasks, with the awareness that hierarchical representations are important (Bengio et al., 2013). Rapid progress in deep learning has shown that mapping and representing topical domains through increasingly abstract layers of feature representation is extremely effective. Unfortunately, this layered representation is difficult to interpret or use for teaching people. Consequently, deep learning models are widely used as algorithmic task performers (e.g. AlphaGo), but few act as theorists or pedagogues. In contrast, our goal is to achieve a deeper-level interpretability that explains not just what has been learned (the end results), but also what is being learned at every single stage (the process).

On the other hand, music theory studies underlying patterns beneath the music surface. It *objectively* reveals higher-level invariances that are hidden from the low-level variations. In practice, the development of music theory is an *empirical* process. Through manual inspection of large corpora of music works, theorists have summarized compositional rules and guidelines (e.g. J. J. Fux, author of *Gradus ad Parnassum*, the most influential book on Renaissance polyphony), and have devised multi-level analytical methods (e.g. H. Schenker, inventor of Schenkerian analysis) to emphasize the hierarchical structure of music, both of which have become the standard materials taught in today's music theory classes. The *objective* and *empirical* nature of music theory suggests the possibility of an *automatic theorist* — statistical techniques that perform hierarchical concept learning — while its *pedagogical* purpose requires human interpretability throughout the entire learning process.

The book title *Gradus ad Parnassum*, means "the path towards Mount Parnassus," the home of poetry, music, and learning. This paper presents MUS-ROVER II, an extension of our prior work (Yu et al., 2016a;b), to independently retake the path towards Parnassus. The rover acts more as a pathfinder than a generative model (e.g. LSTM), emphasizing the *path* more than the *destination*.

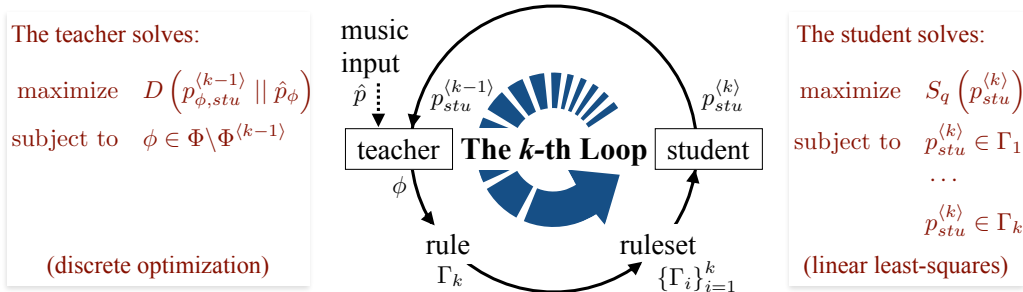

Figure 1: MUS-ROVER's self-learning loop (the $k$th iteration). The teacher (discriminator) takes as inputs the student's latest style $p_{stu}^{\langle k-1 \rangle}$ and the input style $\hat{p}$, and identifies a feature $\phi$ through which the two styles manifest the largest gap $D(\cdot||\cdot)$. The identified feature is then made into a rule (a constraint set $\Gamma_k$), and augments the ruleset $\{\Gamma_i\}_{i=1}^k$. The student (generator) takes as input the augmented ruleset to update its writing style into $p_{stu}^{\langle k \rangle}$, and favors creativity, i.e. more possibilities, by maximizing the Tsallis entropy $S_q$ subject to the rule constraints. In short, the teacher extracts rules while the student applies rules; both perform their tasks by solving optimization problems.

We compare the paths taken by this improved automatic theorist to paths taken by human theorists (say Fux), studying similarities as well as pros and cons of each. So advantages from both can be jointly taken to maximize the utility in music education and research. In this paper in particular, we highlight the concept hierarchy that one would not get from our prior work, as well as enhanced syllabus personalization that one would not typically get from traditional pedagogy.

## 2 MUS-ROVER OVERVIEW

As the first algorithmic pathfinder in music, MUS-ROVER I introduced a "teacher $\rightleftharpoons$ student" model to extract compositional rules for writing 4-part chorales (Yu et al., 2016a;b). The model is implemented by a self-learning loop between a *generative* component (student) and a *discriminative* component (teacher), where both entities cooperate to iterate through the rule-learning process (Figure 1). The student starts as a *tabula rasa* that picks pitches uniformly at random to form sonorities (a generic term for chord) and sonority progressions. The teacher compares the student's writing style (represented by a probabilistic model) with the input style (represented by empirical statistics), identifying one feature per iteration that best reveals the gap between the two styles, and making it a rule for the student to update its probabilistic model. As a result, the student becomes less and less random by obeying more and more rules, and thus, approaches the input style. Collecting from its rule-learning traces, MUS-ROVER I successfully recovered many known rules, such as "Parallel perfect octaves/fifths are rare" and "Tritons are often resolved either inwardly or outwardly".

**What is Inherited from MUS-ROVER I** MUS-ROVER II targets the same goal of learning interpretable music concepts. It inherits the self-learning loop, as well as the following design choices.

(Dataset and Data Representation) We use the same dataset that comprises 370 C scores of Bach's 4-part chorales. We include only pitches and their durations in a piece's *raw* representation, notated as a MIDI matrix whose elements are MIDI numbers for pitches. The matrix preserves the two-dimensional chorale texture, with rows corresponding to melodies, and columns to harmonies.

(Rule Representation) We use the same representation for high-level concepts in terms of rules, unrelated to rules in propositional logic. A (compositional) *rule* is represented by a feature and its distribution: $r = (\phi, p_\phi)$, which describes likelihoods of feature values. It can also be transformed to a linear equality constraint $(A_\phi p_{stu} = p_\phi)$ in the student's optimization problem ($\Gamma$'s in Figure 1).

(Student's Probabilistic Model) We still use $n$-gram models to represent the student's style/belief, with *words* being sonority features, and keep the student's optimization problem as it was. To reiterate the distinctions to many music $n$-grams, we *never* run $n$-grams in the raw feature space, but *only* collectively in the high-level feature spaces to prevent overfitting. So, rules are expressed as probabilistic laws that describe either (vertical) sonority features or their (horizontal) progressions.

**What is New in MUS-ROVER II**   We study *hierarchies* on features, so rules are later presented *not* just as a linear list, but as hierarchical families and sub-families. In particular, we introduce *conceptual hierarchy* that is pre-determined by feature maps, and infer *informational hierarchy* that is post-implied from an information-theoretic perspective. We upgrade the self-learning loop to *adaptively* select memories in a multi-feature multi-$n$-gram language model. This is realized by constructing *hierarchical filters* to filter out conceptual duplicates and informational implications. By further following the information scent spilled by *Bayesian surprise* (Varshney, 2013), the rover can effectively localize the desired features in the feature universe.

## 3   RELATED WORK

**Adversarial or Collaborative**   MUS-ROVER's self-learning loop between the teacher (a discriminator) and student (a generator) shares great *structural* similarity to generative adversarial nets (Goodfellow et al., 2014) and their derivatives (Denton et al., 2015; Makhzani et al., 2015). However, the working mode between the discriminator and generator is different. In current GAN algorithms, the adversarial components are black-boxes to each other, since both are different neural networks that are coupled only end to end. The learned intermediate representation from one model, no matter how expressive or interpretable, is *not* directly shared with the other. Contrarily in MUS-ROVER, both models are transparent to each other (also to us): the student directly leverages the rules from the teacher to update its probabilistic model. In this sense, the learning pair in MUS-ROVER is more *collaborative* rather than *adversarial*. Consequently, not only the learned concepts have interpretations individually, but the entire learning trace is an interpretable, cognitive process.

Furthermore, MUS-ROVER and GAN contrast in the *goal* of learning and the resulting *evaluations*. The rover is neither a classifier nor a density estimator, but rather a pure representation learner that outputs high-level concepts and their hierarchies. Training this type of learner in general is challenging due to the lack of a clear objective or target (Bengio et al., 2013), which drives people to consider some end task like classification and use performance on the task to *indirectly* assess the learned representations. In MUS-ROVER, we introduce information-theoretic criteria to guide the training of the automatic theorist, and in the context of music concept learning, we *directly* evaluate machine generated rules and hierarchies by comparison to those in existing music theory.

**Interpretable Feature Learning**   In the neural network community, much has been done to first recover disentangled representations, and then post-hoc interpret the semantics of the learned features. This line of work includes denoising autoencoders (Vincent et al., 2008) and restricted Boltzmann machines (Hinton et al., 2006; Desjardins et al., 2012), ladder network algorithms (Rasmus et al., 2015), as well as more recent GAN models (Radford et al., 2015). In particular, InfoGAN also introduces information-theoretic criteria to augment the standard GAN cost function, and to some extent achieves interpretability for both discrete and continuous latent factors (Chen et al., 2016). However, beyond the end results, the overall learning process of these neural networks are still far away from human-level concept learning (Lake et al., 2015), so not directly instructional to people.

**Automatic Musicians**   Music *theory* and *composition* form a reciprocal pair, often realized as the complementary cycle of *reduction* and *elaboration* (Laitz, 2016) as walks up and down the multi-level music hierarchy. Accordingly, various models have been introduced to automate this up/down walk, including music generation (Cope & Mayer, 1996; Biles, 1994; Simon et al., 2008), analysis (Taube, 1999), or theory evaluation (Rohrmeier & Cross, 2008). In terms of methodologies, we have rule-based systems (Cope, 1987), language models (Google Brain, 2016; Simon et al., 2008), and information-theoretic approaches (Jacoby et al., 2015; Dubnov & Assayag, 2002). However, all of these models leverage domain knowledge (e.g. human-defined chord types, functions, rules) as part of the model inputs. MUS-ROVER takes as input *only* the raw notations (pitches and durations), and outputs concepts that are comparable to (but also different from) our domain knowledge.

## 4   HIERARCHICAL RULE LEARNING

MUS-ROVER II emphasizes *hierarchy* induction in learning music representations, and divides the induction process into two stages. In the first stage, we impose *conceptual hierarchy* as *pre-defined* structures among candidate features before the self-learning loop. In the second stage, we infer *informational hierarchy* as *post-implied* structures through the rule learning loops.

**Interpretable Features**   A *feature* is a function that computes a distributed representation of the building blocks that constitute data samples. For Bach's 4-part chorales, we model every piece (4-row matrix) as a sequence of sonorities (columns). So every sonority is the building block of its composing piece (like a word in a sentence). Then a feature maps a sonority onto some feature space, summarizing an attribute. To formalize, let $\Omega = \{\texttt{R}, \texttt{p}_1, \ldots, \texttt{p}_n\}$ be an alphabet that comprises a rest symbol $\texttt{R}$, and $n$ pitch symbols $\texttt{p}_i$. In addition, the alphabet symbols — analogous to image pixels — are manipulable by arithmetic operations, such as plus/minus, modulo, and sort. More precisely, every $\texttt{p}_i$ is an integer-valued MIDI number (60 for middle C, granularity 1 for semi-tone), and $\texttt{R}$ is a special character which behaves like a python $\texttt{nan}$ variable. The four coordinates of every sonority $p \in \Omega^4$ denote soprano, alto, tenor, and bass, respectively. We define a *feature* as a surjective function $\phi : \Omega^4 \mapsto \phi(\Omega^4)$, and the corresponding *feature space* by its range. As a first and brutal categorization, we say a feature (space) is *raw* (or *lowest-level*) if $|\phi(\Omega^4)| = |\Omega^4|$, and *high-level* if $|\phi(\Omega^4)| < |\Omega^4|$. For instance, $\Omega^4$ or any permutation of $\Omega^4$ is a raw feature space.

MUS-ROVER II employs a more systematic way of generating the universe of interpretable features. A (sonority) feature is constructed as the composition of a window and a descriptor. A *window* is a function that selects parts of the input sonority: $w_I : \Omega^4 \mapsto \Omega^{|I|}$, where $I$ is an index set. For instance, $w_{\{1,4\}}(p) = (p_1, p_4)$ selects soprano and bass. A *descriptor* is constructed inductively from a set of basis descriptors $B$, consisting of atomic arithmetic operations. We currently set $B = \{\texttt{order}, \texttt{diff}, \texttt{sort}, \texttt{mod}_{12}\}$ (Appendix A.2). We define a descriptor of length $k$ as the composition of $k$ bases: $d_{(k)} = b_k \circ \cdots \circ b_1$, for all $b_i \in B$, where $d_{(0)}$ is the identity function. We collect the family of all possible windows: $W = \{w_I \mid I \in 2^{\{1,2,3,4\}} \setminus \{\emptyset\}\}$, and the family of all descriptors of length less than or equal to $k$: $D^{[k]} = \{d_{(k')} \mid 0 \le k' \le k\}$, and form the *feature universe*:

$$\Phi = \{d \circ w \mid w \in W, d \in D^{[k]}\}. \tag{1}$$

The fact that every candidate feature in $\Phi$ is systematically generated as composition of atomic operators ensures its interpretability, since one can literally read it out step-by-step from the composition.

**Feature-Induced Partition**   On the one hand, a feature function has all the mathematic specifications to name the corresponding feature and feature values. On the other hand, we *only* care about the *partition* of the input domain ($\Omega^4$) induced by the feature but *not* the (superficial) naming of the clusters. In other words, we only identity the sonority clusters whose members are mapped to the same function value, but not the value per se. As a result, we use a partition to refer to the essence of a concept, and the inducing function as a *mathematical name* to interpret the concept. To formalize, a feature function $\phi$ induces a *partition* of its domain

$$\mathcal{P}_\phi = \left\{\phi^{-1}(\{y\}) \mid y \in \phi(\Omega^4)\right\}. \tag{2}$$

Given a feature universe $\Phi$, (2) defines an equivalence relation on $\Phi$: $\phi \overset{\mathcal{P}}{\sim} \phi'$ if $\mathcal{P}_\phi = \mathcal{P}_{\phi'}$, which induces the corresponding partition family $\mathcal{P}_\Phi$ as the resulting equivalence classes. For two partitions $\mathcal{P}, \mathcal{Q} \in \mathcal{P}_\Phi$, we say $\mathcal{P}$ is *finer* than $\mathcal{Q}$ (or $\mathcal{Q}$ is *coarser*), written as $\mathcal{P} \succeq \mathcal{Q}$, if for all $p, p' \in \Omega^4$, $p, p'$ are in the same cluster under $\mathcal{P} \Rightarrow p, p'$ are in the same cluster under $\mathcal{Q}$. We say $\mathcal{P}$ is *strictly finer*, written as $\mathcal{P} \succ \mathcal{Q}$, if $\mathcal{P} \succeq \mathcal{Q}$ and $\mathcal{Q} \not\succeq \mathcal{P}$.

**Conceptual Hierarchy**   Based on the binary relation $\succ$, we construct the *conceptual hierarchy* for the partition family $\mathcal{P}_\Phi$, and represent it as a directed acyclic graph (DAG) with nodes being partitions. For any pair of nodes $v, v'$, $v \to v'$ if and only if the partition referred by $v$ is (strictly) finer than that referred by $v'$. The DAG grows from a single source node, which represents the finest partition — every point in the domain by itself is a cluster — and extends via the edges to coarser and coarser partitions. In terms of features, we say a feature $\phi'$ is at a *higher level* than another feature $\phi$, if the induced partitions satisfy $\mathcal{P}_\phi \succ \mathcal{P}_{\phi'}$. In other words, a higher-level feature induces a coarser partition that ignores lower-level details by merging clusters. One can check that the finest partition (the source node) is indeed induced by a raw feature. We attach an efficient algorithm for pre-computing the conceptual hierarchy in Appendix A.3.

We emphasize the necessity of this multi-step process: features → partitions → hierarchy (DAG), as opposed to a simple hierarchical clustering (tree). The latter loses many inter-connections due to the tree structure and its greedy manner, and more importantly, the interpretability of the partitions.

**Informational Hierarchy**   We infer *informational hierarchy* from a many-to-one relation, called *implication*, along a rule trace. More formally, let $\{r_i\}_{i=1}^k := \{(\phi_i, \hat{p}_{\phi_i})\}_{i=1}^k$ be the extracted trace

of rules (in terms of feature and feature distribution) by the $k$th iteration of the self-learning loop. We say a feature $\phi$ is *informationally implied* from the trace $\{r_i\}_{i=1}^k$ with tolerance $\gamma > 0$, if

$$gap\left(p_{\phi,stu}^{\langle k \rangle} \,\|\, \hat{p}_\phi\right) := D\left(p_{\phi,stu}^{\langle k \rangle} \,\|\, \hat{p}_\phi\right) < \gamma, \quad \text{and} \quad gap\left(p_{\phi,stu}^{\langle k' \rangle} \,\|\, \hat{p}_\phi\right) \geq \gamma, \forall k' < k,$$

where $D(\cdot \| \cdot)$ is the KL divergence used to characterize the gap of the student's style (probabilistic model) against Bach's style (input). One trivial case happens when $\phi$ is extracted as the $k$th rule, i.e. $\phi = \phi_k$, then $gap(p_{\phi',stu}^{\langle k \rangle} \,\|\, \hat{p}_{\phi'}) = 0 < \gamma, \forall \phi' \in \{\phi' \mid \mathcal{P}_\phi \succ \mathcal{P}_{\phi'}\}$, meaning that feature $\phi$, once learned as a rule, informationally implies itself and all its descendants in the conceptual hierarchy. However, what is more interesting is the informational implication from other rules outside the conceptual hierarchy, which is typically hard for humans to "eyeball".

One might question the necessity of conceptual hierarchy since it can be implied in the informational hierarchy. The answer is yes in principle, but no in practice. The main difference is that conceptual hierarchy is pre-computed over the entire feature universe before the loop, which is global, precise, and trace independent. On the contrary, informational hierarchy is trace specific and loose, due to tolerance $\gamma$ and the precision of the optimization solver. As a result, informational hierarchy alone tends to lose the big picture and require more post-hoc interpretations, and is unstable in practice.

**Hierarchical Filters**  Beyond their benefits in revealing inter-relational insights among distributed representations, we build *hierarchical filters* from both conceptual and informational hierarchies, for the purpose of pruning hierarchically entangled features and speeding up feature selection. This upgrades MUS-ROVER II into a more efficient, robust, and cognitive theorist. Recall the skeleton of the teacher's optimization problem in Figure 1, we flesh it out as follows:

$$\underset{\phi \in \Phi}{\text{maximize}} \quad gap\left(p_{\phi,stu}^{\langle k-1 \rangle} \,\|\, \hat{p}_\phi\right) \tag{3}$$

$$\text{subject to} \quad H(\hat{p}_\phi) \leq \delta \qquad \qquad \text{(Regularity Condition)}$$

$$\phi \notin C^{\langle k-1 \rangle} := \left\{\phi \mid \mathcal{P}_\phi \preceq \mathcal{P}_{\phi'}, \phi' \in \Phi^{\langle k-1 \rangle}\right\} \qquad \text{(Conceptual-Hierarchy Filter)}$$

$$\phi \notin I^{\langle k-1 \rangle} := \left\{\phi \mid gap\left(p_{\phi,stu}^{\langle k-1 \rangle} \,\|\, \hat{p}_\phi\right) < \gamma\right\} \qquad \text{(Informational-Hierarchy Filter)}$$

In the above optimization problem, $\Phi$ is the feature universe defined in (1) and $\phi \in \Phi$ is the optimization variable whose optimal value is used to form the $k$th rule: $\phi_k = \phi^\star, r_k = (\phi^\star, \hat{p}_{\phi^\star})$. We decouple the regularity condition from the objective function in our previous work (which was the generalized *cultural hole* function), and state it separately as the first constraint that requires the Shannon entropy of the feature distribution to be no larger than a given threshold (Pape et al., 2015). The second constraint encodes the filter from conceptual hierarchy, which prunes coarser partitions of the learned features $\Phi^{\langle k-1 \rangle} := \{\phi_1, \ldots, \phi_{k-1}\}$. The third constraint encodes the filter from informational hierarchy, which prunes informationally implied features.

There are two hyper-parameters $\delta$ and $\gamma$ in the optimization problem (3), whose detailed usage in syllabus customization will be discussed later in Sec. 6. At a high level, we often pre-select $\gamma$ before the loop to express a user's *satisfaction level*: a smaller $\gamma$ signifies a meticulous user who is harder to satisfy; the threshold $\delta$ upper bounds the *entropic difficulty* of the rules, and is adaptively adjusted through the loop: it starts from a small value (easy rules first), and auto-increases whenever the feasible set of (3) is empty (gradually increases the difficulty when mastering the current level).

## 5  Adaptive Memory Selection

MUS-ROVER II considers a continuous range of higher order $n$-grams (variable memory), and adaptively picks the optimal $n$ based on a balance among multiple criteria. The fact that every $n$-gram is also on multiple high-level feature spaces opens the opportunities for long-term memories without exhausting machine memory, while effectively avoiding overfitting.

**Two-Dimensional Memory**  In light of a continuous range of $n$-grams, say $n \in N = \{2, 3, \ldots\}$, the feature universe adds another dimension, forming a two-dimensional memory ($N \times \Phi$) — *length* versus *depth* — for the language model (Figure 2: left). The length axis enumerates $n$-gram orders, with a longer memory corresponding to a larger $n$; the depth axis enumerates features, with a deeper

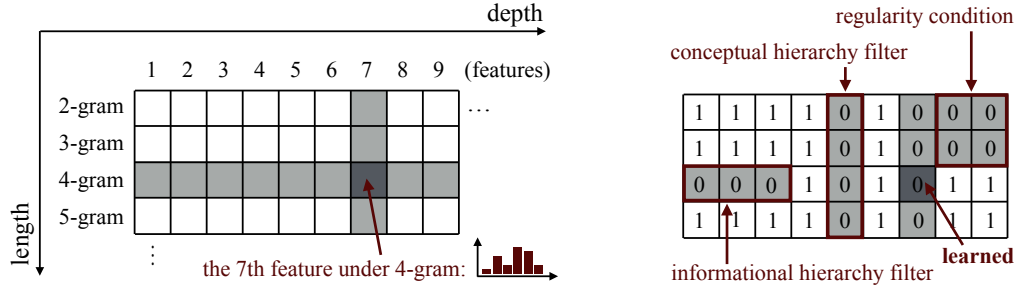

Figure 2: MUS-ROVER II's two-dimensional memory (left): the length axis enumerates $n$-gram orders; the depth axis enumerates features; and every cell is a feature distribution. Memory mask (right): 0 marks the removal of the corresponding cell from feature selection, which is caused by a hierarchical filter or the regularity condition or (contradictory) duplication.

memory corresponding to a higher-level feature. Every cell in the memory is indexed by two coordinates $(n, \phi)$, referring to the feature $\phi$ under the $n$-gram, and stores the corresponding feature distribution. As a consequence, the rule extraction task involves picking the right feature under the right $n$-gram, which extends the space of the optimization problem (3) from $\Phi$ to $N \times \Phi$. Accordingly, the constraints of (3) jointly forge a mask on top of the 2D memory (Figure 2: right).

**Criteria and Balance**  We propose three criteria to extract rules from the 2D memory: confidence, regularity, and efficacy. *Confidence* is quantified by empirical counts: the more relevant examples one sees in Bach's chorales, the more confident. *Regularity* is quantified by Shannon entropy of the rule's feature distribution: a rule is easier to memorize if it is less entropic (Pape et al., 2015). *Efficacy* is inversely quantified by the gap between the student's probabilistic model and the rule's feature distribution: a rule is more effective if it reveals a larger gap. There are tradeoffs among these criteria. For instance, a lower-level feature is usually more effective since it normally reflects larger variations in the gap, but is also unlikely to be regular, thus harder to memorize and generalize. Also a feature under a higher-order $n$-gram may be both regular and effective, but the number of examples that match the long-term conditionals is likely to be small, reducing confidence.

**Adaptive Selection: Follow the (Bayesian) Surprise**  The teacher's optimization problem (3) explicitly expresses the efficacy factor in the objective, and the regularity condition as the first constraint. To further incorporate confidence, we cast the rule's feature distribution $\hat{p}_\phi$ in a Bayesian framework rather than a purely empirical framework as in our previous work. We assume the student's belief with respect to a feature $\phi$ follows a Dirichlet distribution whose expectation is the student's probabilistic model. In the $k$th iteration of the self-learning loop, we set the student's prior belief as the Dirichlet distribution parameterized by the student's latest probabilistic model:

$$\text{prior}_{\phi,stu} \sim \text{Dir}\left(c \cdot p_{\phi,stu}^{\langle k-1 \rangle}\right),$$

where $c > 0$ denotes the strength of the prior. From Bach's chorales, the teacher inspects the empirical counts $q_\phi$ associated with the feature $\phi$ and the relevant $n$-gram, and computes the student's posterior belief if $\phi$ were selected as the rule:

$$\text{posterior}_{\phi,stu} \sim \text{Dir}\left(q_\phi + c \cdot p_{\phi,stu}^{\langle k-1 \rangle}\right).$$

The concentration parameters of the Dirichlet posterior show the balance between empirical counts and the prior. If the total number of empirical counts is small (less confident), the posterior will be smoothed more by the prior, de-emphasizing the empirical distribution from $q_\phi$. If we compute $\hat{p}_\phi \propto \left(q_\phi + c \cdot p_{\phi,stu}^{\langle k-1 \rangle}\right)$ in the objective of (3), then

$$gap\left(p_{\phi,stu}^{\langle k-1 \rangle} \,\|\, \hat{p}_\phi\right) = D\left(\mathbb{E}\left[\text{prior}_{\phi,stu}\right] \,\|\, \mathbb{E}\left[\text{posterior}_{\phi,stu}\right]\right). \tag{4}$$

The right side of (4) is closely related to Bayesian surprise (Varshney, 2013), which takes the form of KL divergence from the prior to posterior. If we remove the expectations and switch the roles between the prior and posterior, we get the exact formula for Bayesian surprise. Both functionals

Table 1: Customizing a syllabus (* signifies rules that are skipped in the faster pace)

| Rule Trace | Faster Pace ($\gamma = 0.5$) | Slower Pace ($\gamma = 0.1$) |
|:---:|:---:|:---:|
| 1 | $\texttt{order} \circ w_{\{1,2,3,4\}}$ | $\texttt{order} \circ w_{\{1,2,3,4\}}$ |
| 2 | $\texttt{mod}_{12} \circ w_{\{1\}}$ | $\texttt{order} \circ \texttt{diff} \circ \texttt{sort} \circ w_{\{1,2,4\}}*$ |
| 3 | $\texttt{mod}_{12} \circ \texttt{diff} \circ w_{\{2,3\}}$ | $\texttt{order} \circ \texttt{diff} \circ \texttt{mod}_{12} \circ w_{\{1,2,3\}}*$ |
| 4 | $\texttt{mod}_{12} \circ \texttt{diff} \circ w_{\{3,4\}}$ | $\texttt{order} \circ \texttt{diff} \circ \texttt{diff} \circ w_{\{1,2,3,4\}}*$ |
| 5 | $\texttt{diff} \circ \texttt{sort} \circ w_{\{2,3\}}$ | $\texttt{order} \circ \texttt{sort} \circ \texttt{mod}_{12} \circ w_{\{2,3,4\}}*$ |
| 6 | $\texttt{mod}_{12} \circ w_{\{3\}}$ | $\texttt{order} \circ \texttt{sort} \circ \texttt{mod}_{12} \circ w_{\{1,3,4\}}*$ |
| 7 | $\texttt{mod}_{12} \circ \texttt{diff} \circ w_{\{1,2\}}$ | $\texttt{order} \circ \texttt{sort} \circ \texttt{mod}_{12} \circ w_{\{1,2,3,4\}}*$ |
| 8 | $\texttt{mod}_{12} \circ \texttt{diff} \circ w_{\{2,4\}}$ | $\texttt{mod}_{12} \circ w_{\{1\}}$ |
| 9 | $\texttt{diff} \circ w_{\{1,2\}}$ | $\texttt{mod}_{12} \circ \texttt{diff} \circ w_{\{2,3\}}$ |
| 10 | $\texttt{diff} \circ \texttt{sort} \circ w_{\{1,3\}}$ | $\texttt{mod}_{12} \circ \texttt{diff} \circ w_{\{3,4\}}$ |

capture the idea of comparing the gap between the prior and posterior. Therefore, the efficacy of concept learning is analogous to seeking (informational) surprise in the learning process.

The subtlety in (4) where we exchange the prior and posterior, makes a distinction from Bayesian surprise due to the asymmetry of KL divergence. As a brief explanation, adopting (4) as the objective tends to produce rules about what Bach hated to do, while the other way produces what Bach liked to do. So we treat it as a design choice and adopt (4), given that rules are often taught as prohibitions (e.g. "parallel fifths/octaves are bad", "never double the tendency tones"). There are more in-depth and information-theoretic discussions on this point (Huszár, 2015; Palomar & Verdú, 2008).

## 6 EXPERIMENTS

MUS-ROVER II's main use case is to produce personalized syllabi that are roadmaps to learning the input style (customized paths to Mount Parnassus). By substituting the student module, users can join the learning cycle, in which they make hands-on compositions and get iterative feedback from the teacher. Alternatively, for faster experimentation, users make the student their learning puppet, which is personalized by its external parameters. This paper discusses the latter case in detail.

**Math-to-Music Dictionary** MUS-ROVER II conceptualizes every rule feature as a partition of the raw space, and uses the inducing function as its mathematical name. To get the meanings of the features, one can simply work out the math, but some of them already have their counterparts as music terminologies. We include a short dictionary of those correspondences in Appendix A.1.

**Pace Control and Syllabus Customization** We present a simple yet flexible pace control panel to the users of MUS-ROVER II, enabling personalized set-up of their learning puppet. The control panel exposes four knobs: the lower bound, upper bound, and stride of the rule's entropic difficulty ($\delta_{min}, \delta_{max}, \delta_{stride}$), as well as the satisfactory gap ($\gamma$). These four hyper-parameters together allow the user to personalize the pace and capacity of her learning experience. The entropic difficulty $\delta$ caps the Shannon entropy of a rule's feature distribution in (3), a surrogate for the complexity (or memorability) of the rule (Pape et al., 2015). It is discretized into a progression staircase from $\delta_{min}$ up to $\delta_{max}$, with incremental $\delta_{stride}$. The resulting syllabus starts with $\delta = \delta_{min}$, the entry level difficulty; and ends whenever $\delta \geq \delta_{max}$, the maximum difficulty that the user can handle. Anywhere in between, the loop deactivates all rules whose difficulties are beyond current $\delta$, and moves onto the next difficulty level $\delta + \delta_{stride}$ if the student's probabilistic model is $\gamma$-close to the input under all currently active rule features.

To showcase syllabus customization, we introduce an ambitious user who demands a faster pace and a patient user who prefers a slower one. In practice, one can collectively tune the stride parameter $\delta_{stride}$ and the gap parameter $\gamma$, with a faster pace corresponding to a larger $\delta_{stride}$ (let's jump directly to the junior year from freshman) and a larger $\gamma$ (having an A- is good enough to move onto the next level, why bother having A+). Here we simply fix $\delta_{stride}$, and let $\gamma$ control the pace. We illustrate two syllabi in Table 1, which compares the first ten (1-gram) rules in a faster ($\gamma = 0.5$) syllabus and a slower one ($\gamma = 0.1$). Notice the faster syllabus gives the fundamentals that a music student will typically learn in her first-year music theory class, including rules on voice crossing,

Table 2: Sample 1-gram rules and their hierarchies.

| | |
|---|---|
| 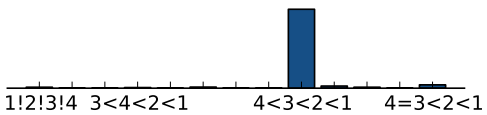 **Interpretable rule (Spacing):** Almost always, the soprano pitch is above the alto, alto above tenor, and tenor above bass. |  This partition sub-family includes 21 coarser partitions, which are local orderings that are already captured by the global ordering. |
|  **Interpretable rule (Scale):** The soprano voice is drawn from a diatonic scale with high probability. | $\mathtt{mod}_{12} \circ w_{\{1\}}$ ● <br> This partition sub-family does not contain any other coarser partitions. |
|  **Interpretable rule (Interval):** The interval of the inner voices are mostly consonant (3,4,5,7,8,9), but perfect octave/unison (0) is rare due to the tight spacing between alto and tenor. | $\mathtt{mod}_{12} \circ \mathtt{diff} \circ w_{\{2,3\}}$ <br> This partition sub-family contains only one coarser partition: <br> $\mathtt{order} \circ \mathtt{sort} \circ \mathtt{mod}_{12} \circ w_{\{2,3\}}$. |
|  **Interpretable rule (Interval):** The interval of the lower voices are mostly consonant, and emerges more perfect octaves due to the wide spacing between tenor and bass. Also, perfect fourth (5) is now considered as a dissonance against the bass. | $\mathtt{mod}_{12} \circ \mathtt{diff} \circ w_{\{3,4\}}$ <br> This partition sub-family contains only one coarser partition: <br> $\mathtt{order} \circ \mathtt{sort} \circ \mathtt{mod}_{12} \circ w_{\{3,4\}}$. |

pitch class set (scale), intervals, and so on (triads and seventh chords will appear later). It effectively skips the nitty-gritty rules (marked by an asterisk) that are learned in the slower setting. Most of these skipped rules do not have direct counterparts in music theory (such as taking the diff operator twice) and are not important, although occasionally the faster syllabus will skip some rules worth mentioning (such as the second rule in the slower pace, which talks about spacing among soprano, alto, and bass). Setting an appropriate pace for a user is important: a pace that is too fast will miss the whole point of knowledge discovery (jump to the low-level details too fast); a pace that is too slow will bury the important points among unimportant ones (hence, lose the big picture).

**Fundamentals: Hierarchical 1-gram** Similar to our teaching of music theory, MUS-ROVER II's proposed syllabus divides into two stages: fundamentals and part writing. The former is under the 1-gram setting, involving knowledge independent of the context; the latter provides online tutoring under multi-$n$-grams. We begin our experiments with fundamentals, and use them to illustrate the two types of feature hierarchies.

Let's take a closer look at the two syllabi in Table 1. The specifications (left) and hierarchies (right) of the four common rules are illustrated in Table 2. The rules' translations are below the corresponding bar charts, all of which are consistent with our music theory. Extracted from the conceptual hierarchy, the right column lists the partition sub-family sourced at each rule, which is pictorially simplified as a tree by hiding implied edges from its corresponding DAG. Every coarser partition in

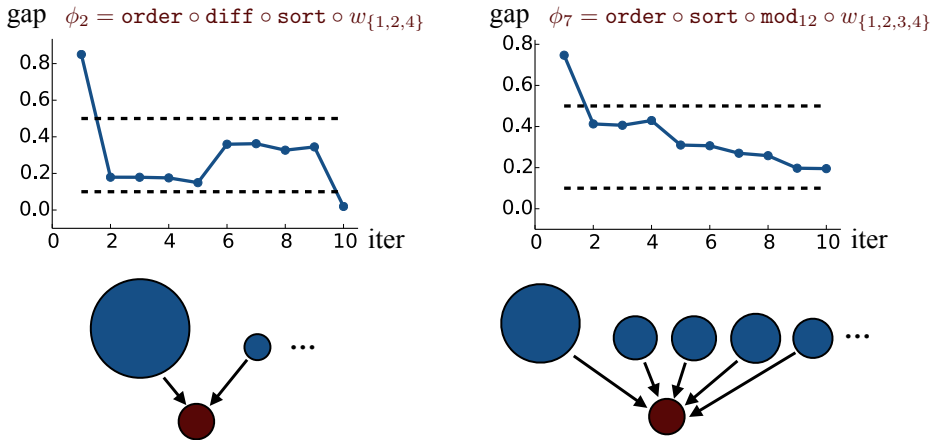

Figure 3: Gap trajectories for two features. The dashed black lines show two different satisfactory gaps ($\gamma = 0.5$ and $0.1$). The bottom charts show the informationally implied hierarchies.

a sub-family is indeed a higher-level representation, but has not accumulated sufficient significance to make itself a rule. A partition will never be learned if one of its finer ancestors has been made a rule. Observe that all of the coarser partitions are not typically taught in theory classes.

MUS-ROVER II measures the student's progress from many different angles in terms of features. With respect to a feature, the gap between the student and Bach is iteratively recorded to form a trajectory when cycling the loop. Studying the *vanishing point* of the trajectory reveals the (local) informational hierarchy around the corresponding feature. Taking the second and seventh rule in the slower syllabus for example, we plot their trajectories in Figure 3. Both illustrate a decreasing trend[1] for gaps in the corresponding feature spaces. The left figure shows that the second rule is largely but not entirely implied by the first, pointing out the hierarchical structure between the two: the first rule may be considered as the dominant ancestor of the second, which is not conceptually apparent, but informationally implied. On the contrary, the right figure shows that the seventh rule is *not* predominantly implied by the first, which instead is informationally connected to many other rules. However, one could say that it is probably safe to skip both rules in light of a faster pace, since they will eventually be learned fairly effectively (with small gaps) but indirectly.

**Part Writing: Adaptive n-grams**  Unlike fundamentals which studies sonority independently along the *vertical* direction of the chorale texture, rules on part writing (e.g. melodic motion, chord progression) are *horizontal*, and *context-dependent*. This naturally results in an online learning framework, in which rule extractions are coupled in the writing process, specific to the realization of a composition (context). Context dependence is captured by the multi-$n$-gram language model, which further leads to the 2D memory pool of features for rule extraction (Sec. 5). Consider an example of online learning and adaptive memory selection, where we have the beginning of a chorale:

$$\langle \mathtt{s} \rangle \to (60, 55, 52, 36) \to (60, 55, 52, 36) \to (62, 59, 55, 43) \to (62, 59, 55, 43) \to (62, 59, 55, 43),$$

and want to learn the probabilistic model for the next sonority. Instead of starting from scratch, MUS-ROVER II launches the self-learning loop with the ruleset initialized by the fundamentals (incremental learning), and considers the 2D memory $N \times \Phi$, for $N = \{2, 3, 4, 5\}$. The first extracted rule is featured by $\mathtt{order} \circ \mathtt{sort} \circ \mathtt{mod}_{12} \circ w_{\{3,4\}}$. The rule is chosen because its corresponding feature has a large confidence level (validated by the large number of matched examples), a small entropy after being smoothed by Bayesian surprise, and reveals a large gap against the Bach's style. Figure 4 shows the relative performance of this rule (in terms of confidence, regularity, and style gap) to other candidate cells in the 2D memory. Among the top 20 rules for this sonority, 12 are 5-gram, 5 are 4-gram, 3 are 2-gram, showing a long and adaptive dependence to preceding context.

**Visualizing Bach's Mind**  With the hierarchical representations in MUS-ROVER II, we are now able to visualize Bach's music mind step by step via activating nodes in the DAG of rule features

---

[1]Fluctuations on the trajectory are largely incurred by the imperfect solver of the optimization problem.

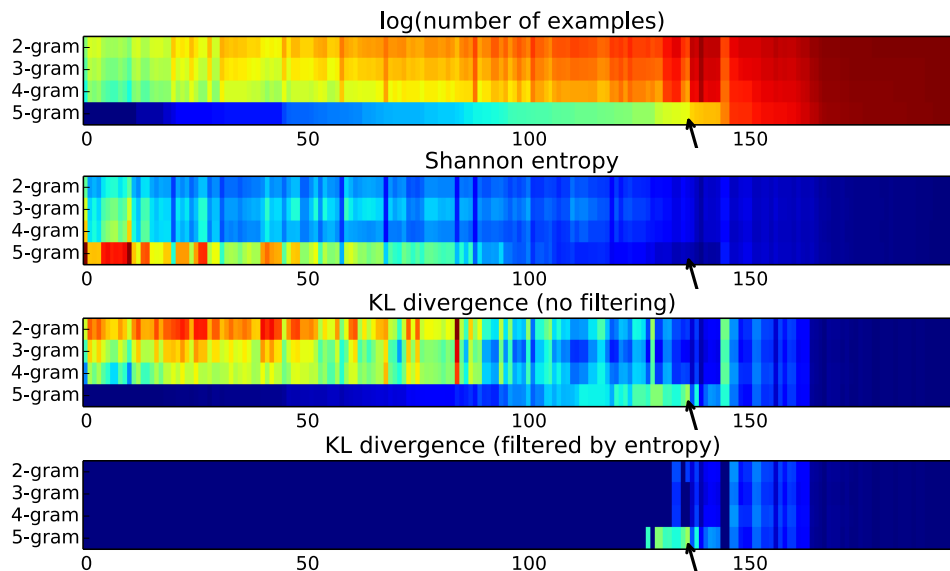

Figure 4: The relative performance of the selected rule (pointed) among the pool of all cells in the 2D memory. A desired rule has: higher confidence (measured by the number of examples, brighter regions in the first row), more regularity (measured by Shannon entropy, darker regions in the second row), and larger style gap (measured by KL divergence, brighter regions in the bottom two rows).

(similar to neuron activations in a brain). The hierarchical structure, as well as the additive activation process, is in stark contrast with the linear sequence of rules extracted from our prior work (Appendix A.5). Figure 5 shows a snapshot of the rule-learning status after ten loops, while the student is writing a sonority in the middle of a piece. The visualization makes it clear how earlier independent rules are now self-organized into sub-families, as well as how rules from a new context overwrite those from an old context, emphasizing that music is highly context-dependent.

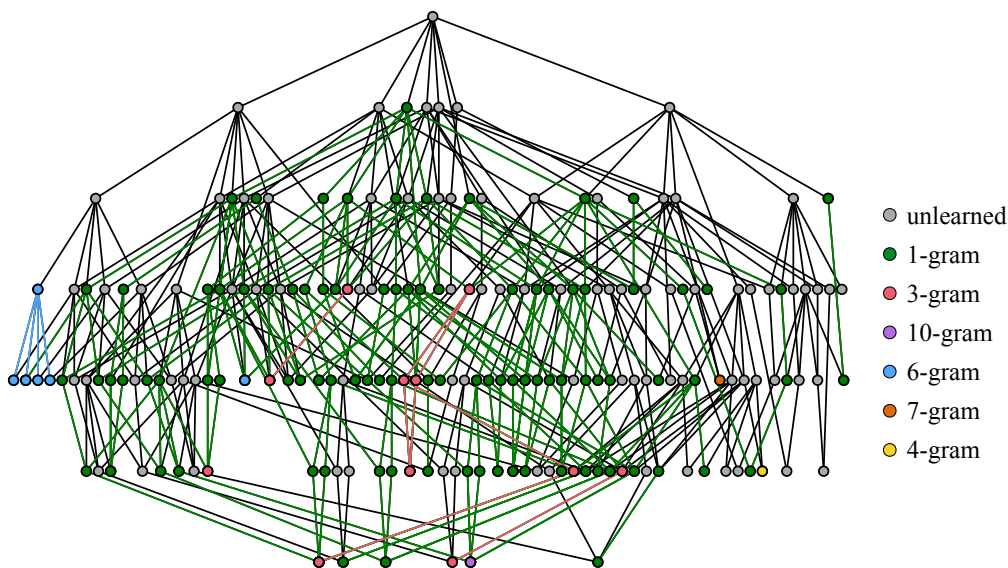

Figure 5: Visualization of Bach's music mind for writing chorales. The underlying DAG represents the conceptual hierarchy (note: edges always point downwards). Colors are used to differentiate rule activations from different $n$-gram settings. We have enlarged $N = \{1, 2, \ldots, 10\}$ to allow even longer-term dependencies.

## 7    CONCLUSIONS AND DISCUSSIONS

Learning hierarchical rules as distributed representations of tonal music has played a central role in music pedagogy for centuries. While our previous work achieved the automation of rule extraction, and to certain level, the interpretability of the rules, this paper yields deeper interpretability that extends to a system of rules and the overall learning process. In summary, it highlights the importance of disentangling the rule features, sorting out their interconnections, and making the concept learning process more dynamic, hierarchical, and cognitive.

MUS-ROVER is targeted to *complement* music teaching and learning. For instance, to many music students, learning and applying rules in part-writing is like learning to solve a puzzle (like Sudoku). Rules themselves are quite flexible as opposed to 0-1 derivatives, and may sometimes be contradictory. In addition, due to the limitation of human short-term memory and the difficulty of foreseeing implications, one has to handle a small set of rules at a time in a *greedy* manner, make some trials, and undo a few steps if no luck. Hence, solving this music puzzle could become a struggle (or maybe interesting): according to personal preferences, one typically begins with a small set of important rules, and via several steps of trial and error, tries one's best to make the part-writing satisfy a majority of rules, with occasional violations on unimportant ones. On the other hand, a machine is often good at solving and learning from puzzles due to its algorithmic nature. For instance, MUS-ROVER's student can take all rules into consideration: load them all at a time as constraints and figure out the *global* optimum of the optimization problem in only a few hours. The same level of efficiency might take a human student years to achieve.

We envision the future of MUS-ROVER as a *partner* to humans in both music teaching and research, which includes but is not limited to, personalizing the learning experience of a student, as well as suggesting new methodologies to music theorists in analyzing and developing new genres. It also has practical applications: as by-products from the self-learning loop, the teacher can be made into a genre classifier, while the student can be cast into a style synthesizer. We are also eager to study the rover's partnership beyond the domain of music.

### ACKNOWLEDGMENTS

We thank Professor Heinrich Taube, President of Illiac Software, Inc., for providing Harmonia's MusicXML corpus of Bach's chorales (`https://harmonia.illiacsoftware.com/`), as well as his helpful comments and suggestions. This work was supported by the IBM-Illinois Center for Cognitive Computing Systems Research (C3SR), a research collaboration as part of the IBM Cognitive Horizons Network.

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

## A  APPENDIX

### A.1  MATH-TO-MUSIC DICTIONARY

Table 3: Sample math-to-music dictionary

| Music Terminology | Feature Map | Feature Values |
|---|---|---|
| pitches in the upper 3 voices | $w_{\{1,2,3\}}$ | $C4 \mapsto 60,\ C\sharp4/D\flat4 \mapsto 61$ |
| pitch class of the bass voice | $\mathtt{mod}_{12} \circ w_{\{4\}}$ | $C \mapsto 0,\ C\sharp/D\flat \mapsto 1$ |
| interval of the inner voices | $\mathtt{diff} \circ w_{\{2,3\}}$ | $P8 \mapsto 12,\ M10 \mapsto 16$ |
| interval class of the outer voices | $\mathtt{mod}_{12} \circ \mathtt{diff} \circ w_{\{1,4\}}$ | $P8 \mapsto 0,\ M10/M3 \mapsto 4$ |
| voicing/spacing | $\mathtt{order} \circ \mathtt{diff} \circ w_I$ | cf. open (closed) position |
| chord regardless of inversion | $\mathtt{sort} \circ \mathtt{mod}_{12} \circ w_I$ | $\mathrm{V}^7/\mathrm{V}_5^6/\mathrm{V}_3^4/\mathrm{V}_2^4 \mapsto (2,4,5,7)$ |
| voice doubling / tripling | $\mathtt{order} \circ \mathtt{sort} \circ \mathtt{mod}_{12} \circ w_I$ | doubling $\mapsto$ "=" |

### A.2  ATOMIC ARITHMETIC OPERATORS

In MUS-ROVER II, we set $B = \{\mathtt{order}, \mathtt{diff}, \mathtt{sort}, \mathtt{mod}_{12}\}$, where

$$\mathtt{diff}(x) = (x_2 - x_1, x_3 - x_2, \cdots), \qquad \forall x \in \Omega^2 \cup \Omega^3 \cup \Omega^4;$$

$$\mathtt{sort}(x) = (x_{(1)}, x_{(2)}, \cdots), \qquad \forall x \in \Omega^2 \cup \Omega^3 \cup \Omega^4;$$

$$\mathtt{mod}_{12}(x) = (\mathrm{mod}(x_1, 12), \mathrm{mod}(x_2, 12), \cdots), \qquad \forall x \in \Omega \cup \Omega^2 \cup \Omega^3 \cup \Omega^4;$$

and $\mathtt{order}(x)$, similar to $\mathtt{argsort}$, maps $x \in \Omega^2 \cup \Omega^3 \cup \Omega^4$ to a string that specifies the ordering of its elements, e.g. $\mathtt{order}((60, 55, 52, 52)) = $ "4=3<2<1". The numbers in an order string denote the indices of the input vector $x$.

### A.3  ALGORITHM FOR CONCEPTUAL HIERARCHY

**Input**:  A family of distinct partitions, represented by a sorted list $\mathtt{P} = [\mathtt{p_1}, \ldots, \mathtt{p_n}]$:
   $\mathtt{p_i} \neq \mathtt{p_j}$, for all $\mathtt{i} \neq \mathtt{j}$, and $|\mathtt{p_1}| \leq \ldots \leq |\mathtt{p_n}|$;
**Output**:  The conceptual hierarchy as a DAG, represented by the $\mathtt{n}$ by $\mathtt{n}$ adjacency matrix $\mathtt{T}$:
   $\mathtt{T[i, j]} = 1$ if there is an edge from node $\mathtt{i}$ to node $\mathtt{j}$ in the DAG;

initialize $\mathtt{T[i, j]} = 0$ for all $\mathtt{i, j}$;
**for** $\mathtt{i = n : 1}$ **do**
    **for** $\mathtt{j = (i + 1) : n}$ **do**
        **if** $\mathtt{T[i, j]} == 0$ **then**
            **if** $\mathtt{is\_coarser(p_i, p_j)}$ **then**
                **for** $\mathtt{k}$ in $\{\mathtt{k} \mid \mathtt{p_j} \prec \mathtt{p_k}\} \cup \{\mathtt{j}\}$ **do**
                    $\mathtt{T[i, k]} = 1$;
                **end**
            **end**
        **end**
    **end**
**end**
$\mathtt{T} = \mathtt{Transpose(T)}$;

**Algorithm 1:** Algorithm for computing the conceptual hierarchy

### A.4  HEURISTICS FOR COMPARING TWO PARTITIONS

Given two partitions $\mathcal{P}, \mathcal{Q}$ from the partition family, the function $\mathtt{is\_coarser}(\mathcal{P}, \mathcal{Q})$ in Algorithm 1 returns $\mathtt{True}$ if $\mathcal{P} \prec \mathcal{Q}$. A brute-force implementation of this function involves studying all (unordered) pairs of elements in the input domain (Hubert & Arabie, 1985), which incurs computational

burdens if the size of the input domain is large. Therefore, we try to get around this brute-force routine whenever certain heuristic can be used to infer the output of `is_coarser` directly. We propose a few of these heuristics as follows.

**Transitivity Heuristic**   If $\mathcal{P}_\phi \succ \mathcal{P}_{\phi'}$ and $\mathcal{P}_{\phi'} \succ \mathcal{P}_{\phi''}$, then $\mathcal{P}_\phi \succ \mathcal{P}_{\phi''}$.

**Window Heuristic**   Let $\mathcal{P}_\phi$ and $\mathcal{P}_{\phi'}$ be two partitions induced by features $\phi$ and $\phi'$, respectively. In addition, $\phi$ and $\phi'$ are generated from the same descriptor that preserves the orders of the inputs' coordinates, e.g. `diff`, `mod`$_{12}$:

$$\phi = d \circ w_I, \quad \phi' = d \circ w_{I'}.$$

We claim that $\mathcal{P}_\phi \succ \mathcal{P}_{\phi'}$, if $I \supset I'$ and $|\phi(\Omega^4)| > |\phi'(\Omega^4)|$. To see why this is the case, pick any $x, y \in \Omega^4$ from the same cluster in $\mathcal{P}_\phi$, then $\phi(x) = \phi(y)$. Since $d$ preserves the orders of the inputs' coordinates, and $I'$ extracts coordinates from $I$, then $\phi'(x) = \phi'(y)$, i.e. $x, y$ are in the same cluster in $\mathcal{P}_{\phi'}$. So, by definition, $\mathcal{P}_\phi \succeq \mathcal{P}_{\phi'}$. Since $|\phi(\Omega^4)| > |\phi'(\Omega^4)|$, $\mathcal{P}_\phi \succ \mathcal{P}_{\phi'}$.

**Descriptor Heuristic**   Let $\mathcal{P}_\phi$ and $\mathcal{P}_{\phi'}$ be two partitions induced by features $\phi$ and $\phi'$, respectively. In addition, $\phi$ and $\phi'$ are generated from the same window:

$$\phi = d \circ w_I; \quad \phi' = d' \circ w_I.$$

We claim that $\mathcal{P}_\phi \succ \mathcal{P}_{\phi'}$, if $d' = b \circ d$ for some function $b$ and $|\phi(\Omega^4)| > |\phi'(\Omega^4)|$. To see why this is the case, pick any $x, y \in \Omega^4$ from the same cluster in $\mathcal{P}_\phi$, then $\phi(x) = \phi(y)$. Since $d' = b \circ d$ for some $b$, then $\phi' = b \circ d \circ w_I = b \circ \phi$, thus, $\phi'(x) = \phi'(y)$, i.e. $x, y$ are in the same cluster in $\mathcal{P}_{\phi'}$. So, by definition, $\mathcal{P}_\phi \succeq \mathcal{P}_{\phi'}$. Since $|\phi(\Omega^4)| > |\phi'(\Omega^4)|$, $\mathcal{P}_\phi \succ \mathcal{P}_{\phi'}$.

**Combined Heuristic**   Combining the above heuristics, one can show that for $\mathcal{P}_\phi$ and $\mathcal{P}_{\phi'}$ where

$$\phi = d \circ w_I; \quad \phi' = d' \circ w_{I'},$$

we have $\mathcal{P}_\phi \succ \mathcal{P}_{\phi'}$, if the following conditions are satisfied:

**1)** $d, d'$ both preserve the orders of the inputs' coordinates,

**2)** $d' = b \circ d$ for some $b$,

**3)** $I \supset I'$,

**4)** $|\phi(\Omega^4)| > |\phi'(\Omega^4)|$.

## A.5   SAMPLE RULE TRACES FROM MUS-ROVER I

Table 4 is essentially the same as Table 2 in our previous publication (Yu et al., 2016a), with feature notations following the current fashion. $\alpha$ is the pace control parameter that we used in our previous system. No hierarchy was present in any of the three rule traces. For instance, the ordering features were learned as independent rules in a trace, even if they are apparently correlated, e.g. the ordering of $w_{\{1,2,3,4\}}$ (S,A,T,B) implies the ordering of $w_{\{1,4\}}$ (S,B).

Table 4: Sample rule traces from MUS-ROVER I

|  | $\alpha = 0.1$ | $\alpha = 0.5$ | $\alpha = 1.0$ |
|---|---|---|---|
| 1 | `order` $\circ\, w_{\{1,4\}}$ | `order` $\circ\, w_{\{1,4\}}$ | $w_{\{1,2,3\}}$ |
| 2 | `order` $\circ\, w_{\{1,3\}}$ | `order` $\circ\, w_{\{1,3\}}$ | $w_{\{2,3,4\}}$ |
| 3 | `order` $\circ\, w_{\{2,4\}}$ | `order` $\circ\, w_{\{2,4\}}$ | `mod`$_{12}$ $\circ\, w_{\{1,2,3,4\}}$ |
| 4 | `order` $\circ\, w_{\{1,2\}}$ | `order` $\circ\, w_{\{1,2\}}$ | $w_{\{1,3,4\}}$ |
| 5 | `order` $\circ\, w_{\{2,3\}}$ | `order` $\circ\, w_{\{2,3,4\}}$ | $w_{\{1,2,4\}}$ |
| 6 | `order` $\circ\, w_{\{3,4\}}$ | $w_{\{1,3,4\}}$ | `diff` $\circ\, w_{\{1,2,3,4\}}$ |
| $\cdots$ | $\cdots$ | $\cdots$ | $\cdots$ |

