# Peer review of "Towards Deep Interpretability (MUS-ROVER II): Learning Hierarchical Representations of Tonal Music"

_ICLR 2017 — accepted_

[Reviewer Comment · AnonReviewer2 · 12 Dec 2016]
**figure clarification**

Do you have any intuition of what is happening in the bottom row (“5-gram”) of the top part of Figure 4?

[Reviewer Comment · AnonReviewer2 · 12 Dec 2016]
**how does the approach scale**

I am curious how well the described approach might scale. Certain aspects (seem to me to) depend on exhaustive searches/enumerations, but this was not entirely clear. Suppose, for example, that rather than Bach chorales, the system was optimized to help students learn orchestration, where there can easily be 10-20 instruments. In this case, the family of windows in Eq(1) would therefore correspond to a significantly larger power set. Furthermore, the pieces are much longer, so counting occurrences of arbitrary compositions of atomic operators could be combinatorially non-trivial, etc. Yet, conceptually organizing the rules of orchestration hierarchically might actually be quite a helpful application. Any comment on this would be appreciated.

[Official Review · AnonReviewer3 · rating 6 · confidence 3 · 16 Dec 2016]
**No Title**

This paper proposes an interesting framework (as a follow-up work of the author's previous paper) to learn compositional rules used to compose better music. The system consists of two components, a generative component (student) and a discriminative component (teacher). The generative component is a Probabilistic Graphical Models, generating the music following learned rules. The teacher compares the generated music with the empirical distribution of exemplar music (e.g, Bach’s chorales) and propose new rules for the student to learn so that it could improve.

The framework is different from GANs that the both the generative and discriminative components are interpretable. From the paper, it seems that the system can indeed learn sensible rules from the composed music and apply them in the next iteration, if trained in a curriculum manner. However, there is no comparison between the proposed system and its previous version, nor comparison between the proposed system and other simple baselines, e.g., an LSTM generative model. This might pose a concern here. 

I found this paper a bit hard to read, partly due to (1) lots of music terms (e.g, Tbl. 1 does not make sense to me) that hinders understanding of how the system performs, and (2) over-complicated math symbols and concept. For example, In Page 4, the concept of raw/high-level feature, Feature-Induced Partition and Conceptual Hierarchy, all means a non-overlapping hierarchical clustering on the 4-dimensional feature space. Also, there seems to be no hierarchy in Informational Hierarchy, but a list of rules. It would be much clearer if the authors write the paper in a plain way. 

Overall, the paper proposes a working system that seems to be interesting. But I am not confident enough to give strong conclusions.

[Official Review · AnonReviewer1 · rating 8 · confidence 4 · 17 Dec 2016]
**No Title**

Summary: 
The paper presents an advanced self-learning model that extracts compositional rules from Bach's chorales, which extends their previous work in: 1) the rule hierarchy in both conceptual and informational dimensions; 2) adaptive 2-D memory selection which assumes the features follow Dirichlet Distribution. Sonority (column of 4 MIDI numbers) acts as word in language model: unigram statistics have been used to learn the fundamental rules in music theory, while n-grams with higher order help characterize part writing. Sonorities have been clustered together based on feature functions through iterations. The partition induced by the features is recognized as a rule if it is sufficiently significant. As a result, two sample syllabi with different difficulty strides and "satisfactory gaps" have been generated in terms of sets of learned rules. 

1. Quality:
 a) Strengths: In the paper, the exploration of hierarchies in two dimensions makes the learning process more cognitive and interpretable. The authors also demonstrate an effective memory selection to speed up the learning.

 b) Flaws: The paper only discussed N<=5, which might limit the learning and interpretation capacities of the proposed model, failing to capture long-distance dependence of music. (In the replies to questions, the authors mentioned they had experimented with max N=10, but I'm not sure why related results were not included in the paper). Besides the elaborated interpretation of results, a survey seeking the opinions of students in a music department might make the evaluation of system performance more persuasive.

2. Clarity:
 a) Pros: The paper clearly delivers an improved automatic theorist system which learns and represents music concepts as well as thoroughly interprets and compares the learned rules with music theory. Proper analogies and examples help the reader perceive the ideas more easily.

 b) Cons: Although detailed definitions can be found in the authors' previous MUS-ROVER I papers, it would be great if they had described the optimization more clearly (in Figure 1. and related parts).  The "(Conceptual-Hierarchy Filter)" row in equations (3): the prime symbol should appear in the subscript.

3. Originality:
The representation of music concepts and rules is still an open area, the paper investigate the topic in a novel way. It illustrates an alternative besides other interpretable feature learning methods such as autoencoders, GAN, etc. 

4. Significance:
It is good to see some corresponding interpretations for the learned rules from music theory. The authors mentioned students in music could and should be involved in the self-learning loop to interact, which is very interesting. I hope their advantages can be combined in the practice of music theory teaching and learning.

[Official Review · AnonReviewer2 · rating 6 · confidence 3 · 21 Dec 2016 (modified: 25 Jan 2017)]

After the discussion below, I looked at previous work by the authors (MUS-ROVER) on which this paper was based. On one hand, this was very helpful for me to better understand the current paper. On the other hand, this was very needed for me to better understand the current paper.

Overall, while I think that I like this work, and while I am familiar with the JSB chorales, with probabilistic approaches, with n-grams, etc, I did find the paper quite hard to follow at various parts. The extensive use of notation did not help the clarity.

I think the ideas and approaches are good, and certainly worth publishing and worth pursuing. I am not sure that, in the paper's current form, ICLR is an appropriate venue. (Incidentally, the issue is not the application as I think that music applications can be very appropriate, nor is the problem necessarily with the approach... see my next suggestion..). I get the sense that a long-form journal publication would actually give the authors the space necessary to fully explain these ideas, provide clearer running examples where needed, provide the necessary background for the appropriate readership, provide the necessary background on the previous system, perhaps demonstrating results on a second dataset to show generality of the approach, etc. A short conference paper just seems to me to be too dense a format for giving this project the description it merits. If it were possible to focus on just one aspect of this system, then that might work, but I do not have good suggestions for exactly how to do that. 

If the paper were revised substantially (though I cannot suggest details for how to do this within the appropriate page count), I would consider raising my score. I do think that the effort would be better invested in turning this into a long (and clearer) journal submission.

[Addendum: based on discussions here & revisions, I have revised my score]

[Author Response · Haizi Yu · 03 Jan 2017]
**Reply to all reviewers**

We thank all the reviewers for their helpful comments and questions!  Below we clarify a few points raised in the reviews and describe the revisions we have made to the paper.

All three reviewers, from different perspectives, suggested better positioning of the paper relative to our own prior work and other existing work. We have taken these comments to heart and revised the paper accordingly. In particular, we have further emphasized the following two connections.

MUS-ROVER II vs. MUS-ROVER I: The purpose of the “MUS-ROVER Overview” section was to clarify the connection between the current rover and its earlier version. We have revised it to make the connection clearer. More specifically, we have updated Figure 1’s caption as well as the “(Rule Representation)” paragraph to include a brief explanation of the student’s optimization problem whose formulation is unchanged in the current rover (Reviewer 1). Also, although we had compared the two rovers in terms of their models, and made it clear what is inherited from our prior work and what is new in our current work, we realize that we did not explicitly compare the two rovers in terms of results and/or contributions. So in the revision, we first state the contribution of our prior work, making it clear that the earlier rover was already able to extract basic voice-leading rules such as “Parallel perfect intervals are rare” (Reviewer 2); and then we emphasize that the current rover not only extracts more rules with longer-term dependencies, but more importantly, studies the hierarchies of the extracted rules. Furthermore, we have added a visualization subsection in the end of the “Experiments” section to further compare the results from the two rovers, i.e. hierarchical rule families and subfamilies vs. unstructured rule sequences (Reviewer 3). In addition, we want to clarify that the purpose of this paper is indeed to focus on one aspect of the system (Reviewer 2) rather than to try to explain everything in details for the entire MUS-ROVER system. The aspect that we emphasize in this paper, is the deeper interpretability that is achieved by rule hierarchies and adaptive 2D memory selection. We totally agree with Reviewer 2 that at some point in the future, a comprehensive journal paper will be the best place to fully explain the entire MUS-ROVER system. However, it is apparent that MUS-ROVER II is not the end of this line of research, and we are making progress in continuing the study of building new versions of rovers, each of which emphasizes a different aspect of the system. It is quite common in our area to publish several conference papers with distinct focuses and then to synthesize them in a longer journal paper to unify the framework. This is the path that we plan to follow.

MUS-ROVER vs. Generative models (e.g. LSTMs): Although MUS-ROVER has a generative component that can be used to generate sample pieces, we do not evaluate the rover’s performance based on its generating power. The reasons are twofold. First, the quality (good or bad) of a music piece is not easy to quantify, though there is emerging work in psychology that suggests the Consensual Assessment Technique (CAT) of evaluation by many experts may be useful. Secondly and more importantly, MUS-ROVER is not targeted as an automatic composer, but as an automatic theorist or pedagogue, so the goal is not to generate pieces, but to explain what has been learned through the entire learning process. We have revised several places in the “Introduction” section to try to make this distinction clear: MUS-ROVER, as a pathfinder to Mount Parnassus, cares about the path towards the destination a lot more than the destination per se. Given that the outputs of generative models (such as LSTMs) are generated samples, they are not comparable to outputs of MUS-ROVER, which are instead ways of generating samples (Reviewer 3).

Reviewer 3 mentioned that the math section on features, feature-induced partitions, and conceptual hierarchy is an over-complicated way of describing non-overlapping hierarchical clustering. We’d like to clarify that it is absolutely necessary to have this multi-step process: features → partitions → hierarchy, and a simple hierarchical clustering wouldn’t work to achieve our goal. The reasons are twofold. 1) Algorithmically, hierarchical clustering will lose many inter-connections due to its greedy algorithm and its tree structure (conceptual hierarchy on the other hand is a DAG). 2) More importantly, hierarchical clustering will lose the interpretability of the resulting partitions: as we pointed out in the “Feature-Induced Partitions” subsection, “We use a partition to refer to the essence of a concept, and the inducing feature function as a mathematical name to interpret the concept”. So without features, having partitions alone will miss the goal of achieving deeper interpretability. To make this clear, we added a small paragraph in the end of the “Conceptual Hierarchy” subsection, emphasizing the necessity of our approach.

Reviewer 3 also mentioned difficulty in understanding the music terms. We apologize if any of the music symbols in the paper cause difficulty for non-musician readers. However, as mentioned above and mentioned in the earlier responses, we do not require readers of this paper to have any music background, and moreover, we do not require the users of MUS-ROVER to have any prior knowledge on music theory either, since the whole purpose of MUS-ROVER is to teach music theory from scratch. The focus of this paper, as opposed to our prior work published in a music venue, is on hierarchies and adaptive 2D memory selection. Taking Table 1 as an example, we do not expect readers to be aware of any underlying music concept so there are no music terms there: all notations are functions, or more precisely, descriptors and windows that are introduced in the “Interpretable Features” subsection. We tried our best to restrict any music-related terms and symbols from the main body of the paper, and put them in the Appendix whenever possible.

Lastly, we strongly agree with Reviewer 1, that deploying MUS-ROVER into a real educational environment will be one of the most exciting things to try next. We are actively collaborating with professors from our music department, to make a live and personalized teaching system in the near future.

[Final Decision · Program Chairs · 06 Feb 2017]
**ICLR committee final decision**

Given that all reviewers were positive aobut this paper and given the unusual application domain, we recommend to accept this paper for poster presentation at the main conference.